**Data Availability Statement:** The data from this study are part of a large multisite consortium and can be made available with use of a data sharing

# Tuberculosis preventive treatment should be considered for all household contacts of pulmonary tuberculosis patients in India

Mandar Paradkar[1,2]*, Chandrasekaran Padmapriyadarsini[3], Divyashri Jain[1], Shri Vijay Bala Yogendra Shivakumar[2], Kannan Thiruvengadam[3], Akshay N. Gupte[4], Beena Thomas[3], Aarti Kinikar[5], Krithika Sekar[3], Renu Bharadwaj[5], Chandra Kumar Dolla[3], Sanjay Gaikwad[5], S. Elilarasi[6], Rahul Lokhande[5], Devarajulu Reddy[3], Lakshmi Murali[3], Vandana Kulkarni[1,2], Neeta Pradhan[1,2], Luke Elizabeth Hanna[3], Sathyamurthi Pattabiraman[3], Rewa Kohli[1,2], Rani S.[3], Nishi Suryavanshi[1,2], Shrinivasa B. M.[3], Samyra R. Cox[4], Sriram Selvaraju[3], Nikhil Gupte[1,2,4], Vidya Mave[1,2,4], Amita Gupta[4], Robert C. Bollinger[4], for the CTRIUMPH-RePORT India Study Team[¶]

1 Byramjee Jeejeebhoy Government Medical College-Johns Hopkins University Clinical Research Site, Pune, Maharashtra, India, 2 Johns Hopkins University Center for Clinical Global Health Education, Pune, Maharashtra, India, 3 National Institute of Research in Tuberculosis, Chennai, Tamil Nadu, India, 4 Johns Hopkins University School of Medicine, Baltimore, Maryland, United States of America, 5 Byramjee Jeejeebhoy Government Medical College and Sassoon General Hospital, Pune, Maharashtra, India, 6 Institute of Child Health and Hospital for Children, Chennai, Tamil Nadu, India

☯ These authors contributed equally to this work.
¶ Members of the CTRIUMPH RePORT India Study team are listed in the Acknowledgments.
* drman23@gmail.com

## Abstract

The World Health Organization (WHO) recently changed its guidance for tuberculosis (TB) preventive treatment (TPT) recommending TPT for all pulmonary TB (PTB) exposed household contacts (HHC) to prevent incident TB disease (iTBD), regardless of TB infection (TBI) status. However, this recommendation was conditional as the strength of evidence was not strong. We assessed risk factors for iTBD in recently-exposed adult and pediatric Indian HHC, to determine which HHC subgroups might benefit most from TPT. We prospectively enrolled consenting HHC of adult PTB patients in Pune and Chennai, India. They underwent clinical, microbiologic and radiologic screening for TB disease (TBD) and TBI, at enrollment, 4–6, 12 and 24 months. TBI testing was performed by tuberculin skin test (TST) and Quantiferon®- Gold-in-Tube (QGIT) assay. HHC without baseline TBD were followed for development of iTBI and iTBD. Using mixed-effect Poisson regression, we assessed baseline characteristics including TBI status, and incident TBI (iTBI) using several TST and/or QGIT cut-offs, as potential risk factors for iTBD. Of 1051 HHC enrolled, 42 (4%) with baseline TBD and 12 (1%) with no baseline TBI test available, were excluded. Of the remaining 997 HHC, 707 (71%) had baseline TBI (TST $\geq$ 5 mm or QGIT $\geq$ 0.35 IU/ml). Overall, 20 HHC (2%) developed iTBD (12 cases/1000 person-years, 95%CI: 8–19). HIV infection (aIRR = 29.08, 95% CI: 2.38–355.77, p = 0.01) and undernutrition (aIRR = 6.16, 95% CI: 1.89–20.03, p = 0.003) were independently associated with iTBD. iTBD was not associated with age, diabetes mellitus, smoking, alcohol, and baseline TBI, or iTBI, regardless of TST ($\geq$ 5 mm, $\geq$ 10 mm, $\geq$ 6 mm increase) or QGIT ($\geq$ 0.35 IU/ml, $\geq$ 0.7 IU/ml) cut-offs. Given the high overall

agreement as per Indian government norms. Specific requests can be placed through: Sameer Khan, Data Manager, BJ Government Medical College Johns Hopkins University Clinical Research Site (sameeriz@hotmail.com) or Robert C Bollinger (rcb@jhmi.edu).

**Funding:** Data in this manuscript were collected as part of the Regional Prospective Observational Research for Tuberculosis (RePORT) India Consortium. CTRIUMPH is part of the RePORT consortium funded with Federal funds from the Government of India's (GOI) Department of Biotechnology (DBT), the Indian Council of Medical Research (ICMR), the USA National Institutes of Health (NIH), the National Institute of Allergy and Infectious Diseases (NIAID), the Office of AIDS Research (OAR), and distributed in part by CRDF Global (USB1-31147-XX-13 to AG). This work was also supported by the National Institutes of Health (NIH 1R01A1097494-01A1 to JG), the NIH funded Johns Hopkins Baltimore-Washington-India Clinical Trials Unit for NIAID Networks (UM1AI069465 to AG, VM, NG). Aarti Kinikar, Rajesh Kulkarni and Rahul Lokhande are supported by the BJGMC JHU HIV TB Program funded by the Fogarty International Center, NIH (D43TW009574 to RCB). The contents of this publication are solely the responsibility of the authors and do not represent the official views of the DBT, the ICMR, the NIH, or CRDF Global. Any mention of trade names, commercial projects or organizations does not imply endorsement by any of the sponsoring organizations. The funders had no role in study design, data collection and analysis, decision to publish, or preparation of the manuscript.

**Competing interests:** The authors have declared that no competing interests exist.

risk of iTBD among recently exposed HHCs, and the lack of association between TBI status and iTBD, our findings support the new WHO recommendation to offer TPT to all HHC of PTB patients residing in a high TB burden country such as India, and do not suggest any benefit of TBI testing at baseline or during follow-up to risk stratify recently-exposed HHC for TPT.

## Introduction

Household contacts (HHC) of pulmonary tuberculosis (PTB) patients are at high risk for acquiring TB infection (TBI) and TB disease (TBD) compared to the general population [1]. Providing TB preventive treatment (TPT) to HHC of PTB patients has been shown to reduce their risk of developing incident TBD (iTBD) [2–5]. In 2018, the World Health Organization (WHO) made new recommendations proposing TPT for all HHC exposed to a patient with PTB, even in high TB prevalence settings, after ruling out active TBD [2]. These recommendations, however, were conditional and made on low quality evidence. Thus, there continues to be a need to assess whether all HHC should be offered TPT and country-specific guidelines vary [6–8]. For example, India is the country with the largest absolute burden of TB accounting for 27% (2.7 of 10 million) cases occurring in the world annually [5]. Indian national guidelines currently recommend limiting TPT to recently-exposed HHC with HIV infection and those <6 years of age, regardless of their tuberculin skin test (TST) status [9–13], due to their increased risk for iTBD [14]. Currently, TPT is not universally recommended for other HHC in resource-constrained settings with high TB burden such as India.

However, in light of the new WHO recommendation, India is currently considering revising their national guidelines. While HIV screening of HHC is recommended in India, screening HHC for other TB risk factors is not incorporated into current national guidelines. Using TST to screen Indian HHC for TBI, as well as screening for a history of diabetes, smoking and/ or alcohol abuse, have been considered to prioritize additional HHC for TPT [2, 15–17]. In addition, while people with TBI have a 5–15% lifetime risk of developing TBD [5], their greatest risk for TBD is within the first 24 months following exposure and their primary TBI diagnosis [18–20]. This suggests that TPT could be particularly beneficial to recently-infected persons of all ages [7]. However, in countries like India with high community rates of TBI in adults of up to 40% [5, 21], it is difficult to determine if older children or adult HHC have recent or more chronic TBI. Therefore, prospectively screening HHC for evidence of TST or QGIT conversion has also been considered, to prioritize offering TPT to HHC with recent TBI.

Routinely screening HHC for the risk factors of iTBD prior to recommending TPT would require significant resources for national TB control programs such as India's. However, there are limited data available about the potential value of additional risk factor screening to inform the national TPT guidelines for TB-exposed HHC [22–24]. Additional relevant challenges include the reliance on TST for TBI diagnosis and the current recommendation to use a TST induration cut-off of ≥10mm when screening HIV-uninfected PTB contacts for TBI [9]. In many other settings, an interferon gamma release assay (IGRA) is also recommended for TBI diagnosis, to determine who is at greater risk for developing TBD and therefore would benefit most from TPT [7]. But, IGRA is not currently recommended for most programs in low- and middle-income settings for diagnosing TBI or for identifying TB-exposed HHC who might benefit from TPT [7, 25].

With India's massive burden of TBD [5], optimizing Indian TPT guidelines for HHC could have a major impact on the global and national TB epidemic [26]. We therefore undertook a study to identify the risk factors for iTBD among Indian adult and pediatric HHC, to determine which HHC might benefit most from TPT.

## Methods

### Study population

Between August 2014 and December 2017, as part of the Cohort for TB Research with Indo-US Medical Partnership (C-TRIUMPH) study [27], the National Institute of Research in Tuberculosis (NIRT), Chennai, and the Byramjee Jeejeebhoy Government Medical College (BJGMC), Pune, India, in an academic collaboration with Johns Hopkins University (JHU), USA, established a cohort of HHC of newly diagnosed PTB patients. The study received human subjects research approvals from- Ethics Committee- BJ Medical College and Sassoon General Hospitals; Institutional Ethics Committee, NIRT; and Johns Hopkins Medicine Institutional Review Board.

A written informed consent was obtained from the participating adult HHC ($\geq$ 18 years of age) and from the legal guardian if the participating HHC was a child <18 years of age. As per the local IRB norms a written informed assent was sought and obtained from children within the age group of $\geq$ 8 to < 18 years. The details of this study design and implementation have been previously described [27–31]. Briefly, HHC were defined as all adults and children living in the same house as an adult ($\geq$18yrs) with active PTB enrolled into C-TRIUMPH, during the 3 months prior to the diagnosis of TBD in this index patient. Index TBD patients were initially diagnosed in local clinics run by the Indian Revised National Tuberculosis Control Program (RNTCP) and then referred to C-TRIUMPH study sites in Chennai and Pune, within one week of index patient's anti-TB treatment initiation. After obtaining informed consent from index PTB patients, study staff approached HHC and those willing to participate in the study were consented and assented as applicable. HHC who did not consent or assent were referred to the RNTCP program, which is responsible for the routine screening of HHC of PTB patients in India.

### Study procedures

At enrollment, we obtained socio-demographics, psycho-social history (tobacco smoking history including the frequency and quantity of smoking, alcohol consumption, and Alcohol Use Disorders Identification Test (AUDIT) score) [32], TB contact history, medical history (current symptoms), history of chronic medical conditions including HIV infection and DM (defined as a known case of DM or, HBA1c $\geq$ 6.5%, or FBG $\geq$ 126 mg/dl or Random Blood glucose $\geq$ 200 mg/dl) and sleep index (whether HHC shared room and bed with index case prior to TBD diagnosis), specimens (sputum smear for AFB, Xpert MTB/Rif, solid and liquid TB cultures, TST, IGRA, HIV, and HbA1c), and chest radiographs. Tobacco smoke use was defined as follows at study entry: never smokers were those who smoked <100 cigarettes in their lifetime and were not current smokers; past smokers were those who smoked $\geq$ 100 cigarettes in their lifetime and were not currently smoking; and current smokers were those who smoked $\geq$100 cigarettes in their lifetime and reported current smoking. Pack years was calculated by multiplying the number of years smoked with the average number of packs (20 smoked tobacco products/pack) per day. Alcohol use disorder (AUD) was defined as having an AUDIT score of at least 8 points [32]. We also conducted a physical examination at baseline, including anthropometry to diagnose undernutrition, which was defined as a composite of body mass index (BMI) < 18.5 kg/m$^2$ for HHC $\geq$ 18 years of age, BMI for age $\leq$ minus 2

standard deviations for HHC >10 and <18 years of age, and weight-for-age ≤ minus 2 standard deviations for HHC ≤10 years of age [33, 34]. At follow-up, psycho-social history, medical history, physical examination and laboratory testing (sputum smear for AFB, Xpert MTB/ Rif, solid and liquid TB cultures, TST and IGRA) were repeated at 4–6, 12 and 24 months. Blood for IGRA testing was collected prior to TST application. If follow-up history or examination revealed signs or symptoms suggestive of active TBD (e.g. history of cough >2 weeks, fever >2 weeks, unexplained weight loss, failure to thrive in children, lymphadenopathy, hepatosplenomegaly, undernutrition), then chest radiograph, sputum for AFB smear, TB cultures, and Xpert MTB/Rif were performed. Additional clinically indicated TB diagnostics were performed for identifying TB in the suspected extrapulmonary sites (e.g., lymph node biopsy for lymphadenopathy, pleural fluid examination for pleural effusion, abdominal ultrasound for pain in abdomen). Index PTB patient data were also collected and analyzed including baseline socio-demographics, psycho-social history, household characteristics (family size, family type, residence type, slum dwelling status, number of windows and family income), other household contact information, medical history, physical examination, laboratory testing (sputum smear for AFB, Xpert MTB/Rif, solid and liquid TB cultures, HIV, and HbA1c), and chest radiographs.

## Diagnosis of TB infection and disease

At follow up, HHC underwent clinical and laboratory evaluations for signs and symptoms of TBD. TST was performed at baseline and repeated at 4–6, 12, and 24 months, if the prior TST was negative (defined as TST <10 mm). TST (Span/Akray diagnostics, India) was administered as 0.1 ml (≤ 5 tuberculin units) of Purified Protein Derivative (PPD) intradermally on the flexor aspect of the forearm. The reaction was read 48–72 hours later. The size of the reaction was determined by measuring the induration diameter in millimeters according to standard published methods [35]. Similarly, QuantiFERON®- TB Gold-in-tube (QGIT, Cellestis, USA) was performed at the baseline and repeated at 4–6, 12, and 24 months, if the prior QGIT was negative (defined as OD <0.35 IU/ml).

**Definitions of TB infection.** TBI was defined by applying each of the three distinct published TST induration cut-offs: ≥5mm [36], an induration increase of >6mm from baseline [37–39], and ≥10mm [38]. Two cut-offs were also applied to define a positive IGRA result, including the standard OD ≥0.35 IU/ml recommended by the manufacturer [40], and an additional cut-off of OD ≥0.70 IU/ml reported to have a higher specificity for TBI in recent studies [41–45]. Two TBI definitions were used for classification of HHC without clinical, microbiological or radiologic evidence of active TBD at baseline:

1. *Prevalent or Baseline TBI of unknown duration included*:

    a. A HHC with a positive TST OR a positive IGRA at baseline OR

    b. A HHC with a positive TST AND a positive IGRA at baseline

2. *Incident TBI (iTBI) included*: Any HHC who was negative for TBD and TBI at baseline, diagnosed to have iTBI using the same criteria that was used at baseline (1a or 1b above) at any subsequent follow-up visit.

**Definitions of TB disease.** An HHC was defined as having microbiologically confirmed TBD if they had a specimen from any source (e.g., sputum, CSF, lymph node) that was positive by TB culture or GeneXpert/MTB Rif. An HHC was defined as having probable TBD if all specimens were negative by TB culture and GeneXpert, but they had a specimen from any source (e.g., sputum, CSF, lymph node) that was positive on AFB smear. An HHC was defined

as having possible TBD if all specimens were negative by TB culture, GeneXpert and AFB, but they were treated empirically for TB by the RNTCP based on clinical and/or radiologic findings. These same criteria were used to classify HHC as having prevalent TBD (defined as TBD diagnosed at enrollment) or iTBD (defined as TB diagnosed at any follow-up assessments among those who had no TBD at enrollment) [9].

## Statistical analysis

Our analyses of HHC at baseline, included additional data that was not available for earlier C-TRIUMPH publications [29, 30]. After excluding HHC found to have active TBD at initial screening, baseline characteristics, including characteristics of the TB index patients and households, of the asymptomatic HHC with and without TBD were compared. Since prior studies have suggested that TB contacts with evidence of TBI should be prioritized for TPT, TST and IGRA results of the HHC at the baseline were analyzed. All p-values were two-sided with statistical significance evaluated at the 0.05 alpha level.

Next, the person-time follow up was calculated as the duration between date of enrollment of HHC to last study follow up date for each HHC, who were at risk of the event (iTBD or iTBI) being considered. The iTBD rate was calculated as number of HHC developing iTBD, per 1000 person-years (PY) follow up. To determine if the iTBD rate was dependent on the baseline TBI definition that was applied, iTBD rates for various combinations of TST cut offs ($\geq$5mm, $\geq$10mm, increase of $\geq$6mm) and IGRA cut offs ($\geq$0.35, $\geq$0.7 IU/ml), were compared. To describe the timing of TBD occurrence we calculated the proportion of HHC with TBD that were diagnosed at each study timepoint.

To determine if screening for specific risk factors could identify HHC at higher risk for progression to TBD that could be prioritized for TPT, we examined the association between baseline characteristics of HHC (including multiple TST and/or IGRA definitions of baseline TBI) and iTBD, using univariate, multivariable, and mixed-effect Poisson regression analyses. The HHC characteristics found to be associated with iTBD in the univariate analysis were included in the overall model and/or the adult multivariate models, as relevant. Additionally, those HHC characteristics that were not statistically significant in the univariate analysis but known to be the published risk factors for iTBD [2, 15–17], were included in the multivariate model. Next, to determine if evidence of recent TBI among the HHC was associated with an increased risk for iTBD, the iTBI rate was calculated as number of HHC developing iTBI, per 1000 PY follow up. To determine if the iTBI rate among HHC was dependent on the TBI definition applied, iTBI rates using various combinations of baseline and follow-up TST cut offs ($\geq$5mm, $\geq$10mm, increase of $\geq$6mm) and IGRA cut offs ($\geq$0.35, $\geq$0.7 IU/ml) were calculated and compared. We then compared iTBD rates for HHC with and without evidence iTBI, for each of the iTBI definitions.

## Results

### Characteristics of HHC, index PTB patients and households

As shown in **Fig 1**, a total of 1051 HHC from 442 households were enrolled. Forty-two (4%) HHC with TBD at baseline and 12 (1.1%) HHC with no baseline TBI testing available, were excluded. The median time from enrollment of the PTB index case to enrollment of their HHC was 1 day (Interquartile range (IQR): 0 to 8 days). PTB index patients reported an average symptom duration of 1.5 months (IQR: 1.0 to 3.0 months), prior to their TBD diagnosis and treatment initiation. HHC found to have baseline TBD reported an average symptom duration of 7 days (IQR: 3 to 8 days). Our analyses included 997 (95%) HHC with no evidence of baseline TBD and at least one TBI test available.

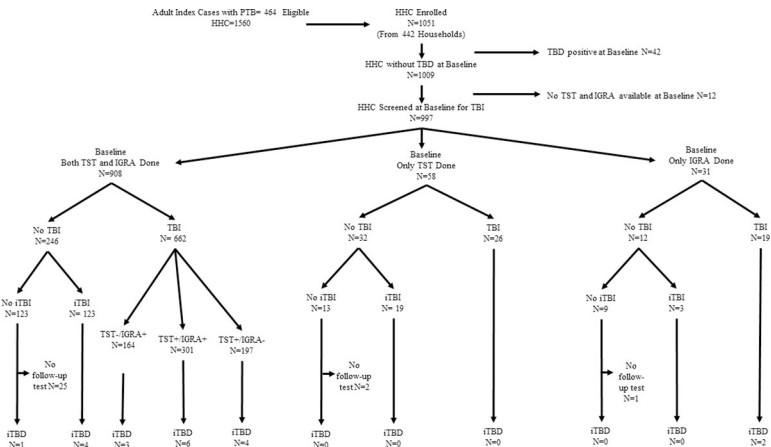

**Fig 1. Screening of household contacts of adult pulmonary TB patients in India.** Flowchart depicts the screening of household contacts (HHC) of adult pulmonary TB (PTB) patients in India, and shows that 1051 HHC enrolled in the study, 997 with no baseline TB disease (TBD) and with at least one baseline test for TB infection (TBI) available, were included in the final analysis. These 997 asymptomatic HHC were classified as those with and without baseline TBI (baseline TBI was defined as TST $\geq$ 5 mm and/or IGRA $\geq$ 0.35 IU/ml). HHC without baseline TBI were further classified as those with and without incident TBI (iTBI was defined as TST $\geq$ 5 mm and/or IGRA $\geq$ 0.35 IU/ml at follow up). Finally, the number of HHC who developed Incident TB disease (iTBD) was stratified by their TBI status.

As shown in **Table 1**, most (68%) HHC were adults $\geq$18 years of age, 56% were female, 2% were HIV-infected, 70% had a BCG scar, and 4% had past history of TB. Among adult HHC, 9% had DM, 9% were current smoker, 3% prior smokers, and 20% consumed alcohol.

The PTB index patients for these HHC were predominantly male (61%), under the age of 45 years (63%), 92% had microbiologically confirmed TBD, 7% were HIV-infected, 26% had DM, 13% were current smokers, 18% former smokers and 44% consumed alcohol (**Table 2**). The average family size for the HHC was 5 members (IQR: 4 to 6), 69% were urban households and 71% of HHC slept in the same room as the PTB index patient (**Table 3**).

Of 997 HHC, 908 (91%) had both TST and IGRA performed, 58 (6%) had only TST and 31 (3%) had only IGRA test performed. Seven-hundred-seven (71%) HHC had evidence of TBI with at least one test positive at baseline screening, of whom 301 (43%) demonstrated both a TST $\geq$5 mm and an IGRA OD $\geq$0.35 IU/ml. Of 290 (29%) HHC with no evidence of TBI at baseline, 246 (85%) had both tests performed and demonstrated a TST <5 mm and an IGRA OD <0.35 IU/ml while the remaining 44 (15%) HHC had only one of these two tests performed with a negative result (**Fig 1**).

### Rates and risk factors for incident TB disease

Of 997 HHC, 20 (2%) subsequently developed iTBD during the 24-month follow-up period, yielding an estimated overall iTBD rate of 12 per 1000 PY (95% CI = 8 to 19/1000 PY). Of these, 2 HHC were under 6 years of age, yielding an estimated iTBD rate of 15 per 1000 PY (95% CI = 2 to 53/1000 PY). **Fig 2** and **S1 Table** show the iTBD rates stratified by different definitions of baseline TBI. The median time from enrollment of the PTB index patient to development of iTBD in the HHC was 5.6 months (Range: 4.4 to 10.4 months) with 13/20 (65%) occurring within 6 months of their baseline screening.

Baseline characteristics of HHC, including their TBI status, were analyzed to identify risk factors for subsequent TBD that might identify HHC in India that could be prioritized for TPT. The univariate analyses identified only bed sharing with the index PTB patient (IRR-3.60, 95%CI: 0.99 to 13.07, p = 0.05) as a risk factor for iTBD (**Tables 1–3**). A multivariate

**Table 1. Baseline characteristics of household contacts and risk factors for incident TB disease among household contacts of adult pulmonary TB patients in India.**

| HHC Characteristics | Overall | iTBD | IR/1000 PY (95% CI) | Univariate IRR (95% CI) | p-value | Adjusted IRR (95% CI) (Overall model) | p-value |
|---|---|---|---|---|---|---|---|
| **Denominators** | **997** | **20** | | | | | |
| Age group (years) | | | | | | 0.99 (0.96–1.03)[a] | 0.78 |
| < 6 | 83 (8%) | 2 (10%) | 15 (2–53) | 1 | | | |
| 6–12 | 144 (14%) | 4 (20%) | 17 (5–44) | 1.17 (0.21–6.37) | 0.86 | | |
| 13–17 | 94 (9%) | 0 | 0 | NA | NA | | |
| 18–44 | 500 (50%) | 11 (55%) | 14 (7–24) | 0.92 (0.20–4.17) | 0.92 | | |
| ≥45 | 176 (18%) | 3 (15%) | 10 (2–30) | 0.70 (0.12–4.21) | 0.7 | | |
| Gender | | | | | | | |
| Male | 442 (44%) | 10 (50%) | 14 (7–25) | 1.20 (0.50–2.88) | 0.69 | 0.86 (0.25–2.91) | 0.80 |
| Female | 555 (56%) | 10 (50%) | 11 (5–21) | 1 | | 1 | |
| Marital status | | | | | | Not included | |
| Never married | 189 (26%) | 5 (36%) | 16 (5–38) | 1 | | | |
| Married | 498 (69%) | 9 (64%) | 11 (5–21) | 0.69 (0.23–2.07) | 0.51 | | |
| Divorced/ Widowed | 34 (5%) | 0 | 0 | NA | | | |
| Level of education | | | | | | Not included | |
| ≤ Primary | 351 (37%) | 9 (47%) | 16 (7–30) | 1 | | | |
| Highschool | 399 (42%) | 6 (32%) | 9 (3–20) | 0.58 (0.21–1.62) | 0.3 | | |
| > Junior college | 199 (21%) | 4 (21%) | 13 (3–33) | 0.80 (0.25–2.6) | 0.71 | | |
| HIV infection | | | | | | | |
| Positive | 15 (2%) | 2 (11%) | 99 (12–359) | 3.20 (0.93–10.99) | 0.06 | 29.08 (2.38–355.77) | 0.01 |
| Negative | 937 (98%) | 17 (89%) | 111 (7–179) | 1 | | 1 | |
| Baseline DM[b] | | | | | | Not included | |
| Yes | 67 (9%) | 1 (7%) | 8 (0–47) | 0.64 (0.21–1.91) | 0.42 | | |
| No | 667 (91%) | 13 (93%) | 12 (7–21) | 1 | | | |
| Incident DM[b] | | | | | | Not included | |
| Yes | 3 (0%) | 0 | 0 | NA | | | |
| No | 992 (100%) | 20 (100%) | 13 (8–19) | | | | |
| Smoking[c] | | | | | | | |
| Current | 58 (9%) | 0 | 0 | NA | 0.46 | Not included | |
| Former | 21 (3%) | 1 (8%) | 27 (1–149) | 2.15 (0.28–16.57) | | | |
| Never | 597 (88%) | 12 (92%) | 12 (6–22) | 1 | | | |
| Alcohol consumption[d] | | | | | | Not included | |
| Yes | 129 (13%) | 3 (15%) | 14 (3–41) | 1.09 (0.30–3.91) | 0.89 | | |
| No | 530 (54%) | 11 (55%) | 13 (6–23) | 1 | | | |
| BCG scar | | | | | | Not included | |
| Present | 569 (70%) | 10 (71%) | 11 (5–20) | 1.09 (0.34–3.48) | 0.88 | | |
| Absent | 240 (30%) | 4 (29%) | 10 (3–25) | 1 | | | |
| Past history of TB | | | | | | Not included | |
| Yes | 36 (4%) | 0 | 0 | NA | | | |
| No | 959 (96%) | 20 (100%) | 13 (8–20) | | | | |
| Undernourished[e] | | | | | | | |
| No | 788 (81%) | 13 (70%) | 10 (5–17) | 1 | | 1 | |
| Yes | 187 (19%) | 7 (30%) | 23 (9–48) | 2.30 (0.92–5.76) | 0.08 | 6.16 (1.89–20.03) | 0.003 |
| Baseline TST[f] | | | | | | Not included | |
| Negative | 714 (74%) | 13 (72%) | 11 (6–19) | 1 | | | |

*(Continued)*

**Table 1.** (Continued)

| HHC Characteristics | Overall | iTBD | IR/1000 PY (95% CI) | Univariate IRR (95% CI) | p-value | Adjusted IRR (95% CI) (Overall model) | p-value |
|---|---|---|---|---|---|---|---|
| Positive (≥10mm) | 252 (26%) | 5 (28%) | 13 (4–30) | 1.17 (0.42–3.27) | 0.77 | | |
| Baseline TST[f] | | | | | | Not included | |
| Negative | 442 (44%) | 8 (44%) | 11 (5–22) | 1 | | | |
| Positive (≥5mm) | 524 (56%) | 10 (56%) | 12 (6–22) | 1.08 (0.43–2.74) | 0.87 | | |
| Baseline IGRA[g] | | | | | | Not included | |
| Negative | 455 (48%) | 9 (47%) | 12 (6–23) | 1 | | | |
| Positive | 484 (52%) | 11 (53%) | 14 (7–26) | 1.17 (0.48–2.81) | 0.73 | | |
| Baseline TST[f] & IGRA[g] | | | | | | Not included* | |
| Both Negative | 246 (45%) | 5 (45%) | 14 (7–27) | 1 | | | |
| Both positive (TST≥5mm) | 301 (55%) | 6 (55%) | 18 (6–42) | 1.01 (0.31–3.29) | >0.95 | | |
| Baseline TST[f] &/or IGRA[g] | | | | | | | |
| Both Negative | 290 (29%) | 5 (25%) | 11 (3–25) | 1 | | 1 | |
| Either Positive (TST≥5mm) | 707 (71%) | 15 (75%) | 13 (7–22) | 1.27 (0.46–3.48) | 0.65 | 1.68 (0.45–6.30) | 0.44 |
| Baseline TST[f] & IGRA[g] | | | | | | Not included* | |
| Both Negative | 183 (32%) | 5 (36%) | 14 (7–27) | 1 | | | |
| Both positive (TST≥10mm) | 386 (68%) | 9 (64%) | 18 (6–42) | 1.26 (0.42–3.77) | 0.67 | | |
| Baseline TST[f] &/or IGRA[g] | | | | | | Not included* | |
| Both Negative | 444 (45%) | 9 (47%) | 13 (6–24) | 1 | | | |
| Either Positive (TST≥10mm) | 553 (55%) | 11 (53%) | 12 (6–22) | 0.99 (0.41–2.41) | >0.95 | | |

HHC = Household contacts of adult patients with pulmonary TB; TBD = TB disease; SD = standard deviation

a. Age is used as linear variable in multivariable models

b. Diabetes mellitus (DM): defined as- Known case of DM or, HB A1c ≥ 6.5%, or Fasting Blood Glucose (FBG) ≥ 126 mg/dl or Random Blood glucose ≥ 200 mg/dl

c. Smoking: Never smokers defined as those HHC who smoked <100 cigarettes in their lifetime and were not current smokers; Past smokers defined as those HHC who smoked ≥ 100 cigarettes in their lifetime and were not currently smoking; and Current smokers defined as those HHC who smoked ≥100 cigarettes in their lifetime and reported current smoking

d. Alcohol: defined as history of any consumption of alcoholic drink

e. Undernourished: defined as BMI < 18.5 kg/m² for HHC ≥ 18 years of age, BMI for age ≤ minus 2 SD for HHC >10 years and <18 years of age and weight for age ≤ minus 2 SD for HHC ≤10 years of age

f. TST- positive test defined as an induration of ≥ 5 mm or more

g. IGRA- positive test defined as ≥0.35 IU/ml.

model that included HHC characteristics (age, gender, HIV infection status, nutritional status, and baseline TBI status), index patient characteristics (age, gender, HIV infection status, sputum smear and culture status, and household characteristics (residence type, family income, and sleep index); identified HIV infection in HHC (aIRR = 29.08, 95%CI: 2.38 to 355.77, p = 0.01) and undernutrition (aIRR = 6.16, 95%CI: 1.89 to 20.03, p = 0.003) to be independently associated with increased risk of iTBD. Interestingly, there was no statistically significant difference in the risk of iTBD between HHC with and without baseline TBI, regardless of TST (≥5mm,≥10mm, ≥6mm increase) or QGIT (≥0.35, ≥0.7IU/L) cut-offs used to define baseline TBI or any other factors. A similar adult-only model that included DM, smoking and alcohol consumption in HHC as risk factors, both HIV (aIRR = 31.57, 95%CI: 0.98 to 1020.42, p = 0.05) and undernutrition (aIRR = 9.88, 95%CI: 2.07 to 47.11, p = 0.004) were independently associated with iTBD (Data not shown). These two risk factors however only accounted for 8 (40%) of the 20 iTBD cases (2 with HIV-infection and 7 with undernutrition).

**Table 2. Baseline characteristics of pulmonary TB index patients and risk factors for incident TB disease among household contacts of adult pulmonary TB patients in India.**

| Index Case Characteristics | Overall | iTBD | IR/1000 PY (95% CI) | Univariate IRR (95% CI) | p-value | Adjusted IRR (95% CI) (Overall model) | p-value |
|---|---|---|---|---|---|---|---|
| **Denominators** | **997** | **20** | | | | | |
| Age group (years) | | | | | | | |
| 18–44 | 633 (63%) | 12 (60%) | 12 (6–21) | 1 | | 1 | |
| ≥45 | 364 (37%) | 8 (40%) | 13 (6–26) | 1.07 (0.44–2.61) | 0.88 | 2.63 (0.80–8.64) | 0.11 |
| Gender | | | | | | | |
| Male | 611 (61%) | 11 (55%) | 11 (6–20) | 0.77 (0.32–1.86) | 0.56 | 2.50 (0.63–9.84) | 0.19 |
| Female | 386 (39%) | 9 (45%) | 15 (7–28) | 1 | | 1 | |
| HIV infection | | | | | | | |
| Positive | 66 (7%) | 3 (16%) | 34 (7–100) | 3.20 (0.93–10.99) | 0.06 | 0.47 (0.04–5.19) | 0.53 |
| Negative | 923 (93%) | 16 (84%) | 11 (6–17) | 1 | | 1 | |
| Baseline Diabetes Mellitus[a] | | | | | | Not included | |
| Yes | 259 (26%) | 4 (20%) | 9 (2–23) | 0.64 (0.21–1.91) | 0.42 | | |
| No | 733 (74%) | 16 (80%) | 14 (8–23) | 1 | | | |
| Smoking[b] | | | | | | Not included | |
| Current | 127 (13%) | 3 (15%) | 15 (3–45) | 1.04 (0.30–3.56) | 0.95 | | |
| Former | 184 (18%) | 1 (5%) | 3 (0–18) | 0.22 (0.03–1.63) | 0.14 | | |
| Non-Smoker | 684 (69%) | 16 (80%) | 15 (8–24) | 1 | | | |
| Alcohol consumption[c] | | | | | | Not included | |
| Yes | 430 (44%) | 6 (30%) | 8 (3–18) | 0.53 (0.20–1.38) | 0.2 | | |
| No | 555 (56%) | 14 (70%) | 16 (9–27) | 1 | | | |
| Microbiologically confirmed TB[d] | | | | | | Not included | |
| Yes | 895 (92%) | 18 (90%) | 12 (7–20) | 0.70 (0.16–3.01) | 0.63 | | |
| No | 83 (8%) | 2 (10%) | 18 (2–64) | 1 | | | |
| Xpert MTB/Rif | | | | | | Not included | |
| Positive | 709 (85%) | 14 (78%) | 12 (7–21) | 0.54 (0.18–1.63) | 0.27 | | |
| Negative | 124 (15%) | 4 (22%) | 23 (6–59) | 1 | | | |
| Smear[e]/Culture[f] (Baseline) | | | | | | | |
| Culture+/Smear- | 135 (15%) | 1 (5%) | 5 (1–29) | 0.46 (0.04–5.02) | 0.52 | NA | |
| Culture-/Smear+ | 12 (1%) | 1 (5%) | 46 (1–257) | 3.98 (0.36–43.87) | 0.26 | 11.50 (0.71–186.51) | 0.09 |
| Culture+/Smear+ | 664 (72%) | 16 (80%) | 15 (8–24) | 1.26 (0.29–5.50) | 0.75 | 1.34 (0.20–8.87) | 0.76 |
| Culture-/Smear- | 113 (12%) | 2 (10%) | 12 (1–42) | 1 | | 1 | |
| Smear[e]/Culture[f] (Month 1) | | | | | | Not included | |

*(Continued)*

**Table 2.** (Continued)

| Index Case Characteristics | Overall | iTBD | IR/1000 PY (95% CI) | Univariate IRR (95% CI) | p-value | Adjusted IRR (95% CI) (Overall model) | p-value |
|---|---|---|---|---|---|---|---|
| Culture+/Smear- | 244 (28%) | 4 (21%) | 10 (3–26) | 0.52 (0.17–1.65) | 0.26 | | |
| Culture-/Smear+ | 44 (5%) | 0 | 0 | NA | NA | | |
| Culture+/Smear+ | 184 (21%) | 3 (16%) | 9 (2–27) | 0.48 (0.14–1.71) | 0.26 | | |
| Culture-/Smear- | 401 (46%) | 12 (63%) | 19 (1–34) | 1 | | | |
| Smear[e]/Culture[f] (Month 2) | | | | | | Not included | |
| Culture+/Smear- | 109 (13%) | 4 (25%) | 22 (6–57) | 1.87 (0.60–5.79) | 0.28 | | |
| Culture-/Smear+ | 53 (6%) | 0 | 0 | NA | NA | | |
| Culture+/Smear+ | 55 (7%) | 0 | 0 | NA | NA | | |
| Culture-/Smear- | 604 (74%) | 12 (75%) | 12 (6–21) | 1 | | | |
| AFB (Baseline) | | | | | | Not included | |
| Negative | 418 (42%) | 8 (40%) | 13 (5–25) | 1 | | | |
| Positive | 572 (58%) | 12 (60%) | 13 (6–22) | 0.99 (0.41–2.43) | >0.95 | | |
| LJ Colony count (Baseline) (CFU/HPF) | | | | | | Not included | |
| Negative | 214 (22%) | 7 (37%) | 21 (8–43) | 1 | | | |
| < 10 | 99 (10%) | 2 (11%) | 12 (2–45) | 0.60 (0.12–2.87) | 0.52 | | |
| 10–100 | 350 (36%) | 4 (21%) | 8 (2–20) | 0.37 (0.11–1.27) | 0.12 | | |
| > 100 | 306 (32%) | 6 (32%) | 11 (4–24) | 0.52 (0.18–1.55) | 0.24 | | |
| Index cavitory disease | | | | | | Not included | |
| No | 382 (43%) | 7 (41%) | 11 (4–23) | 1 | | | |
| Yes | 507 (57%) | 10 (59%) | 12 (6–22) | 1.09 (0.41–2.85) | 0.87 | | |

TBD = TB disease; CFU/HPF = Colony Forming Units per High Power Field

a. Diabetes mellitus (DM): defined as- Known case of DM or, HB A1c $\geq$ 6.5%, or Fasting Blood Glucose (FBG) $\geq$ 126 mg/dl or Random Blood glucose $\geq$ 200 mg/dl

b. Current smoker: Never smokers defined as those HHC who smoked <100 cigarettes in their lifetime and were not current smokers; Past smokers defined as those HHC who smoked $\geq$ 100 cigarettes in their lifetime and were not currently smoking; and Current smokers defined as those HHC who smoked $\geq$100 cigarettes in their lifetime and reported current smoking

c. Alcohol: defined as history of any consumption of alcoholic drink

d. AFB smear and/or Gene Xpert and/or TB culture positive

e. AFB Smear positive defined as scanty, 1+, 2+ or 3+ result

f. Culture positive defined as any culture (MGIT &/or LJ) positive for MTB

### TB infection conversion and risk of incident TB disease

We further examined if recent conversion of TB infection (i.e. iTBI) was associated with risk of iTBD by first calculating the rates of TST and IGRA conversion. Two-hundred twenty-one (89.8%) of 246 HHC who were TBI negative for both tests at baseline had at least 1 subsequent follow-up test available. Of these 221, 123 (55.7%) HHC converted their TBI status, with an

**Table 3. Baseline characteristics of the household and risk factors for incident TB disease among household contacts of adult pulmonary TB patients in India.**

| Household Characteristics | Overall | iTBD | IR/1000 PY (95% CI) | Univariate IRR (95% CI) | p-value | Adjusted IRR (95% CI) (Overall model) | p-value |
|---|---|---|---|---|---|---|---|
| Denominators | 997 | 20 | | | | | |
| Family size | 5 (4–6) | 5 (4.5–7) | NA | 1.09 (0.94–1.26) | 0.25 | Not included | |
| Family type | | | | | | Not included | |
| Nuclear | 647 (65%) | 13 (65%) | 12 (7–21) | 1 | | | |
| Joint | 350 (35%) | 7 (35%) | 13 (5–26) | 1.02 (0.41–2.56) | >0.95 | | |
| Residing in slum | | | | | | Not included | |
| Yes | 262 (27%) | 7 (35%) | 18 (7–37) | 1.65(0.66–4.14) | 0.29 | | |
| No | 716 (73%) | 13 (65%) | 11 (6–19) | 1 | | | |
| Residence type | | | | | | | |
| Urban | 687 (69%) | 17 (85%) | 16 (9–26) | 2.86 (0.84–9.75) | 0.09 | 5.03 (1.02–24.75) | 0.05 |
| Rural | 310 (31%) | 3 (15%) | 6 (1–16) | 1 | | 1 | |
| Windows (median) | 2 (1–3) | 2 (1–2.5) | NA | 0.85 (0.64–1.12) | 0.25 | Not included | |
| Family income (Rupees) | | | | | | | |
| <10000 | 412 (43%) | 8 (53%) | 12 (5–24) | 1 | | 1 | |
| 10000–30000 | 510 (53%) | 6 (40%) | 7 (3–16) | 0.58 (0.20–1.67) | 0.31 | 0.90 (0.28–2.90) | 0.85 |
| >30000 | 33 (4%) | 1 (7%) | 21 (1–118) | 1.72 (0.22–13.77) | 0.61 | 5.16 (0.39–68.62) | 0.21 |
| Sleep index[a] | | | | | | | |
| Different room | 276 (30%) | 3 (15%) | 6 (1–18) | 1 | | 1 | |
| Same room/same bed | 397 (42%) | 10 (50%) | 23 (11–42) | 3.60 (0.99–13.07) | 0.05 | 2.10 (0.38–11.56) | 0.39 |
| same room/different bed | 276 (29%) | 7 (35%) | 11 (5–23) | 1.78 (0.46–6.89) | 0.4 | 0.69 (0.12–3.97) | 0.68 |

TBD = TB disease.

a. Sleep location of adult pulmonary TB index patient.

overall iTBI estimate of 491 per 1000 PY (95% CI 408 to 586/1000 PY). Of 123 HHC with iTBI, 67 (82%) were detected within 6 months of baseline screening. iTBI estimates varied depending on the TBI definition used (**Fig 3** **and** **S2 Table**).

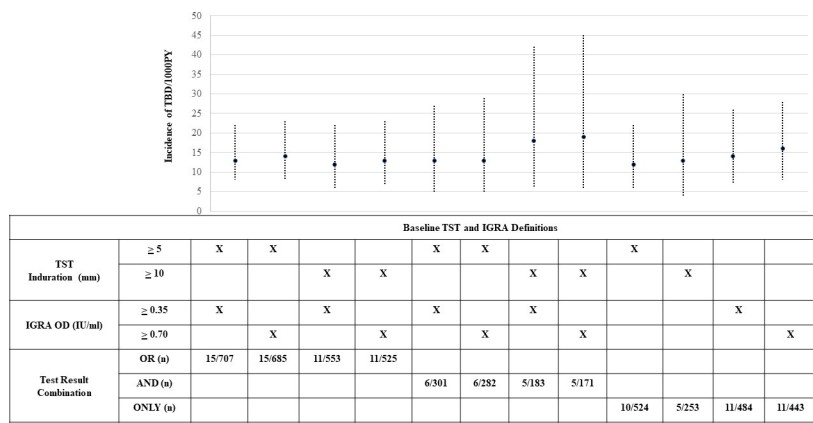

**Fig 2. Incidence of TB disease among household contacts of adult pulmonary TB patients in India, stratified by TST and/or IGRA cut offs used to define baseline TB infection.** Line graph depicting the incident TB disease (iTBD) rates among household contacts (HHC) of adult pulmonary TB (PTB) patients in India, stratified using different TST and/or IGRA cut offs to define the baseline TB infection (TBI) status. The graph shows that the iTBD rates were similar irrespective of the individual test cut off used to define baseline TBI, and irrespective of whether the definition used both a positive TST and IGRA test ("AND") or either test alone ("OR").

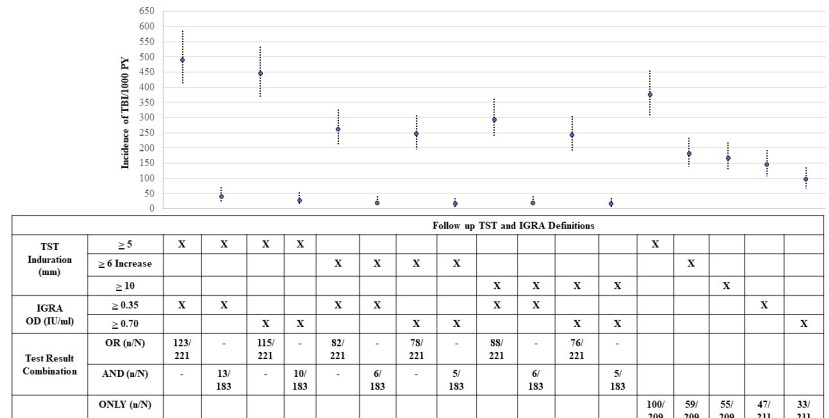

| Follow up TST and IGRA Definitions | | | | | | | | | | | | | | | |
|---|---|---|---|---|---|---|---|---|---|---|---|---|---|---|---|
| **TST Induration (mm)** | ≥ 5 | X | X | X | X | | | | | | | | X | | | |
| | ≥ 6 Increase | | | | | X | X | X | X | | | | | X | | |
| | ≥ 10 | | | | | | | | | X | X | X | X | | X | |
| **IGRA OD (IU/ml)** | ≥ 0.35 | X | X | | | X | X | | | X | X | | | | | X |
| | ≥ 0.70 | | | X | X | | | X | X | | | X | X | | | X |
| **Test Result Combination** | OR (n/N) | 123/221 | - | 115/221 | - | 82/221 | - | 78/221 | - | 88/221 | - | 76/221 | - | | | |
| | AND (n/N) | - | 13/183 | - | 10/183 | - | 6/183 | - | 5/183 | - | 6/183 | - | 5/183 | | | |
| | ONLY (n/N) | | | | | | | | | | | | | 100/209 | 59/209 | 55/209 | 47/211 | 33/211 |

**Fig 3. Incidence of TB infection among household contacts of adult pulmonary TB patients in India, stratified by baseline TST and/or IGRA cut offs.** Line graph depicting the incident TB infection (iTBI) rates among household contacts (HHC) of adult pulmonary TB (PTB) patients in India, stratified using different thresholds for TST conversion and/or IGRA conversion to define iTBI status. The graph shows the rates of iTBI were higher for lower TST/IGRA conversion thresholds as compared to higher TST/IGRA conversion; and the iTBI rates were higher when the iTBI definition used either test positive ("OR") criteria as compared to both test positive ("AND") criteria. In addition, changing the cut-off for a positive IGRA from ≥0.35 to ≥ 0.70 IU/ml did not significantly impact the iTBI estimates. However, increasing the induration TST cut-off from ≥ 5 mm to ≥ 10 mm or requiring a ≥ 6 mm increase in induration from previous reading resulted in lower iTBI estimates, regardless of the IGRA cut off used.

Finally, we compared and found that the iTBD estimates were similar for those HHC with and without incident TBI, irrespective of the TST or IGRA cutoffs used to define iTBI. In addition, the iTBD estimates were similar irrespective of whether the definition of iTBI was based on the requirement of both a positive TST and IGRA test ("AND") or either test alone ("OR") (**Fig 4; S3 Table**).

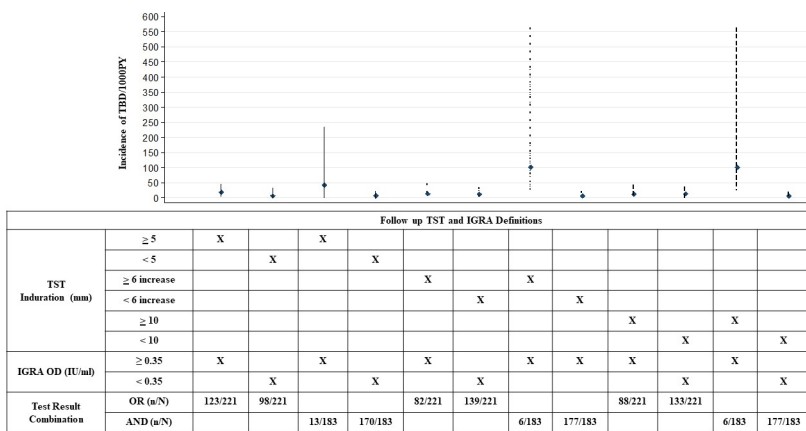

| Follow up TST and IGRA Definitions | | | | | | | | | | |
|---|---|---|---|---|---|---|---|---|---|---|
| **TST Induration (mm)** | ≥ 5 | X | | X | | | | | | |
| | < 5 | | X | | X | | | | | |
| | ≥ 6 increase | | | | | X | | X | | |
| | < 6 increase | | | | | | X | | X | |
| | ≥ 10 | | | | | | | | X | X |
| | < 10 | | | | | | | | | X | X |
| **IGRA OD (IU/ml)** | ≥ 0.35 | X | | X | | X | | X | X | X | X |
| | < 0.35 | | X | | X | | X | | | X | | X |
| **Test Result Combination** | OR (n/N) | 123/221 | 98/221 | | | 82/221 | 139/221 | | 88/221 | 133/221 | |
| | AND (n/N) | | | 13/183 | 170/183 | | | 6/183 | 177/183 | | 6/183 | 177/183 |

**Fig 4. Incident TB disease rates among household contacts with and without incident TB infection.** A line graph depicting the incident TB disease (iTBD) rates among household contacts (HHC) of adult pulmonary TB (PTB) patients in India, stratified by incident TB infection (iTBI) status which was defined using different TST conversion cut offs (≥ 5 mm, ≥ 10 mm, and ≥ 6 mm increase in induration from previous reading) along with the IGRA conversion cut off of ≥ 0.35 IU/l. The comparison shows that the iTBD estimates were similar for those HHC with and without iTBI, irrespective of the TST or IGRA cutoffs used to define iTBI. In addition, the iTBD estimates were similar irrespective of whether the definition of iTBI was based on the requirement of both a positive TST and IGRA test ("AND") or either test alone ("OR").

## Discussion

Our study includes a number of findings that could inform national guidelines for the screening and provision of TPT for HHC of PTB patients in India, the country that accounts for 27% of all global TBD. We found that 68% of the HHC diagnosed with TBD and 71% HHCs diagnosed with TBI were detected at baseline. This highlights that households of PTB patients are hot spots for TB transmission. Furthermore, our study reports the first robust estimates of both iTBI and iTBD rates, among HHC of PTB patients from India, both of which were very high. Finally, we found that most HHC characteristics including baseline TBI status and evidence of recent TBI (iTBI) were not associated with an increased risk for iTBD. Given the high baseline risk of iTBD in HHC, and the lack of association between TBI status and iTBD, our results support the new WHO guidelines, to offer TPT to all HHC of PTB patients in India.,.

This study and our previous publications have demonstrated high (4%) prevalence of baseline TBD at time of initial HHC screening [28, 31], which is similar to prior studies from India [46, 47], as well as pooled estimates from meta-analyses from low- and middle-income countries [1]. Our findings confirm what has been shown in other countries [48]—timely screening of HHC is an important active TB case finding strategy for India and should be a high priority for the RNTCP. Our finding that 71% of HHC without TBD already had evidence of TBI at initial screening suggests that most iTBI in HHC occurred before HHC screening is initiated in India and is consistent with prior studies [29, 49]. The WHO now recommends TPT for all adults and pediatric HHC exposed to an adult patient with PTB after ruling out the active TBD. [2] However, the Indian national guidelines currently recommend limiting TPT to HHC with HIV and those <6 years of age, regardless of their TST status [9]. Our study, demonstrates that the risk for TBD remains very high among Indian HHC who have not already developed active TBD at the time of initial contact screening. Our iTBD rate of 12 per 1000 PY in HHC of index PTB patients is similar to recently reported iTBD rates from South Africa (13 per 1000 PY) [50], and Peru (9.3 per 1000 PY) [51]. Our iTBD rates are higher than the annual iTBD rates in the general population of India (1.99 per 1000 people) and higher than the average rate reported from other high TB burden countries (1.8 per 1000 people), including South Africa (5.2 per 1000 people) [5].

We found that only baseline HIV and undernutrition were independently associated with iTBD. No other baseline HHC characteristics including age, gender, DM, smoking, alcohol, baseline TBI and iTBI status, were associated with a higher risk of developing iTBD. These findings were confirmed independently in two distinct multivariate analytical models- an overall model including all the age groups, and a model restricted to the adult population which additionally included variables for DM, smoking (pack-years based quantification of tobacco smoking) and alcohol consumption. However, none of these were found to be independent risk factors for iTBD in either of the analytical model. This may be partly explained based on a published literature review stating that importance of these risk factors depends on prevalence of each risk factor and therefore is subject to the variations between regions and countries [15]. Furthermore, our analysis involved clustered HHCs and high-risk contacts were likely already diagnosed with prevalent TBD at enrollment and therefore excluded from the analysis. While some of the HHC subgroups like those with undernutrition and HIV infection had higher risk of iTBD and may benefit the most from TPT, the overall iTBD risk in HHC was much higher than what has been reported in the general population in India. This finding is in line with the WHO guidelines stating that TBI testing and risk categorization by DM status, nutritional status, smoking or alcohol use is not recommended for TPT initiation [8].

Other studies have reported that the greatest risk for TBD among HHC is within the first 12–24 months of their primary TBI [18–20], suggesting that they might have the greatest

benefit of TPT. Our study also provides the first estimates of iTBI among HHC, based on TST and/or QGIT conversion from India. We defined the iTBI status in multiple possible ways, using various published TST and/or QGIT conversion cut offs, both in combinations of TST and IGRA and also by the individual test result. However, we found that irrespective of the iTBI definition used, iTBI was also not a risk factor for iTBD among HHC within the first 2 years of follow-up. This suggests that there is little utility in identifying and prioritizing HHC for TPT who have iTBI. Thus, our data suggest that in the absence of a diagnostic test that can more accurately predict who will develop iTBD, all HHC should be offered TPT.

An important limitation of our study was that we could not assess the impact of TPT (INH prophylaxis) in preventing the progression to TBD, because it is not currently recommended for all adult HHC in India. We also could not assess the predictive value of sustained versus transient IGRA conversion, with the risk of progression to TBD in different age groups [52].

In summary, our study supports the new WHO guidelines to rapidly screen all HHC of PTB patients and to offer TPT to all HHC without TBD and do not suggest any clear benefit of TBI testing at baseline or during follow-up to further risk stratify recently-exposed HHC for targeted TPT [7].

## Supporting information

**S1 Table. Incidence rates for TB disease among household contacts of adult pulmonary TB patients in India.** This table shows the incidence of TB disease (iTBD) among the household contacts (HHC) of adult pulmonary TB (PTB) patients in India, stratified using different TST ($\geq 5$ mm, $\geq 10$ mm) and/or IGRA ($\geq 0.35$ IU/ml, $\geq 0.7$ IU/ml) cut offs to define the baseline TB infection (TBI) status. The iTBD rates were similar irrespective of the individual test cut off used to define baseline TBI, and irrespective of whether the definition used both a positive TST and IGRA test ("AND") or either test alone ("OR").
(DOCX)

**S2 Table. Incidence rates for TB infection among household contacts of adult pulmonary TB patients in India.** This table shows the incident TB infection (iTBI) rates among household contacts (HHC) of adult pulmonary TB (PTB) patients in India, stratified using different thresholds for TST conversion and/or IGRA conversion to define iTBI status. The rates of iTBI were higher for lower TST/IGRA conversion thresholds as compared to higher TST/IGRA conversion; and the iTBI rates were higher when the iTBI definition used either test positive ("OR") criteria as compared to both test positive ("AND") criteria. In addition, changing the cut-off for a positive IGRA from $\geq 0.35$ to $\geq 0.70$ IU/ml did not significantly impact the iTBI estimates. However, increasing the induration TST cut-off from $\geq 5$ mm to $\geq 10$ mm or requiring a $\geq 6$ mm increase in induration from previous reading resulted in lower iTBI estimates, regardless of the IGRA cut off used.
(DOCX)

**S3 Table. Incident rates for TB disease among household contacts with and without incident TB infection.** This table shows the incident TB disease (iTBD) rates among household contacts (HHC) of adult pulmonary TB (PTB) patients in India, stratified by incident TB infection (iTBI) status which was defined using different TST conversion cut offs ($\geq 5$ mm, $\geq 10$ mm, and $\geq 6$ mm increase in induration from previous reading) along with the IGRA conversion cut off of $\geq 0.35$ IU/l. The comparison shows that the iTBD estimates were similar for those HHC with and without iTBI, irrespective of the TST or IGRA cutoffs used to define iTBI. In addition, the iTBD estimates were similar irrespective of whether the definition of iTBI was based on the requirement of both a positive TST and IGRA test ("AND") or either

test alone ("OR").
(DOCX)

## Acknowledgments

The CTRIUMPH-RePORT India study team (Lead author: Amita Gupta) listed in alphabetical order—Aarti Kinikar[5], Akshay N Gupte[4], Amita Gupta[4], Amita Nagraj[1,2], Anand Kumar B[3], Andrea DeLuca[4], Anita More[1], Anju Kagal[5], Archana Gaikwad[1], Ashwini Nangude[1], Ayesha Momin[1], Balaji S[3], Beena Thomas[3], Bency Joseph[3], Bharath TK[3], Brindha B[3], Chhaya Valvi[5], Chandrasekaran Padmapriyadarsini[3], Deepak Pole[1], Deepanjali Biradar[1], Devanathan A[3], Devarajulu Reddy[3], Devi Sangamithrai M[3], Dhanaji Jagdale[1], Dileep Kadam[5], Divyashri Jain[1], Dolla CK[3], S.Elilarasi[6], Gabriela Smit[4], Gangadarsharma R[3], Geetha Ramachandran[3], Hanumant Chaugule[1], Hari Koli[1], Hemanth Kumar[3], Jeeva J[3], Jessica Elf[4], Jonathan Golub[4], Jyoti Chandane[1], Kannan M[3], Kannan Thiruvengadam[3], Karthikesh M[3], Karunakaran S[3], Kelly Dooley[4], Lakshmi Murali[3], Lavanya M[3], Luke Hanna[3], Madasamy S[3], Madeshwaran A[3], Mageshkumar M[3], Mangaiyarkarasi S[3], Mahesh Gujare[1], Mandar Paradkar[1,2], Manoharan S[3], Michel Premkumar M[3], Mrunalini Kamble[1], Munivardhan P[3], Murugesan S[3], Gomathy NS[3], Neeta Pradhan[1,2], Nikhil Gupte[1,2,4], Nishi Suryavanshi[1,2], Ponnuraja C[3], Poonam Patil[1], Prasad Deshpande[1], Prasanna Sahoo[1], Pratik Awale[1], Premkumar N[3], Rahul Lokhande[5], Rajkumar S[3], Ranganathan K[3], Rani S[3], Rani V[3], Renu Bharadwaj[5], Renu Madewar[1], Rengaraj R[3], Rewa Kohli[1,2], Robert Bollinger[4], Rosemarie Warlick[4], Rupak Shivakoti[4], Sahadev Javanjal[1], Samir Shaikh[1], Samyra R. Cox[4], Sandhya Khadse[5], Sanjay Gaikwad[5], Sathyamurthi P[3], Savita Kanade[1,2], Shalini Pawar[1], Shashank Hande[1], Shital Muley[1], Shital Sali[1], Shri Vijay Bala Yogendra Shivakumar[2], Shrinivasa BM[3], Shyam Biswal[3], Silambu Chelvi K[3], Smita Nimkar[1,2], Soumya Swaminathan[3], Sriram Selvaraju[3], Suba priya K[3], Sunanda Kamble[1], Sundeep Salvi[7], Sushant Meshram[5], Surendhar S[3], Swapnil Raskar[1], Swapnali Lakare[1], Syed Hissar[3], Uma Devi[3], Vandana Kulkarni[1,2], Vidula Hulyalkar[1], Vidya Mave[1,2,4], Vinod Taywade[1], Vrinda Bansode[1], Yogesh Daware[1], Zaheda Khan[1]

1 Byramjee Jeejeebhoy Government Medical College-Johns Hopkins University Clinical Research Site, Pune, Maharashtra, India

2 Johns Hopkins University Center for Clinical Global Health Education, Pune Office, Maharashtra, India

3 National Institute of Research in Tuberculosis, Chennai, Tamil Nadu, India

4 Johns Hopkins University School of Medicine, Baltimore, Maryland, United States of America

5 Byramjee Jeejeebhoy Government Medical College and Sassoon General Hospital, Pune, Maharashtra, India

6 Institute of Child Health and Hospital for Children, Chennai, Tamil Nadu, India

7 Chest Research Foundation, Pune, India

## Author Contributions

**Conceptualization:** Mandar Paradkar, Chandrasekaran Padmapriyadarsini, Nikhil Gupte, Vidya Mave, Amita Gupta, Robert C. Bollinger.

**Data curation:** Divyashri Jain, Shri Vijay Bala Yogendra Shivakumar, Kannan Thiruvengadam, Akshay N. Gupte, Nikhil Gupte, Robert C. Bollinger.

**Formal analysis:** Divyashri Jain, Kannan Thiruvengadam, Nikhil Gupte.

**Funding acquisition:** Chandrasekaran Padmapriyadarsini, Samyra R. Cox, Vidya Mave, Amita Gupta.

**Investigation:** Mandar Paradkar, Chandrasekaran Padmapriyadarsini, Shri Vijay Bala Yogendra Shivakumar, Kannan Thiruvengadam, Beena Thomas, Aarti Kinikar, Krithika Sekar, Renu Bharadwaj, Chandra Kumar Dolla, Sanjay Gaikwad, S. Elilarasi, Rahul Lokhande, Devarajulu Reddy, Lakshmi Murali, Vandana Kulkarni, Neeta Pradhan, Luke Elizabeth Hanna, Sathyamurthi Pattabiraman, Rewa Kohli, Rani S., Nishi Suryavanshi, Shrinivasa B. M., Sriram Selvaraju, Nikhil Gupte, Vidya Mave.

**Methodology:** Mandar Paradkar, Chandrasekaran Padmapriyadarsini, Shri Vijay Bala Yogendra Shivakumar, Kannan Thiruvengadam, Akshay N. Gupte, Beena Thomas, Krithika Sekar, Renu Bharadwaj, S. Elilarasi, Vandana Kulkarni, Neeta Pradhan, Luke Elizabeth Hanna, Sathyamurthi Pattabiraman, Rewa Kohli, Rani S., Nishi Suryavanshi, Shrinivasa B. M., Sriram Selvaraju, Nikhil Gupte, Vidya Mave, Amita Gupta, Robert C. Bollinger.

**Project administration:** Mandar Paradkar, Chandrasekaran Padmapriyadarsini, Shri Vijay Bala Yogendra Shivakumar, Krithika Sekar, Renu Bharadwaj, Vandana Kulkarni, Luke Elizabeth Hanna, Rewa Kohli, Nishi Suryavanshi, Samyra R. Cox, Nikhil Gupte, Vidya Mave, Amita Gupta.

**Resources:** Samyra R. Cox, Vidya Mave, Amita Gupta.

**Supervision:** Mandar Paradkar, Chandrasekaran Padmapriyadarsini, Shri Vijay Bala Yogendra Shivakumar, Akshay N. Gupte, Beena Thomas, Aarti Kinikar, Renu Bharadwaj, Sanjay Gaikwad, Rahul Lokhande, Lakshmi Murali, Vandana Kulkarni, Neeta Pradhan, Luke Elizabeth Hanna, Nishi Suryavanshi, Nikhil Gupte, Vidya Mave, Amita Gupta.

**Visualization:** Divyashri Jain.

**Writing – original draft:** Mandar Paradkar, Amita Gupta, Robert C. Bollinger.

**Writing – review & editing:** Mandar Paradkar, Chandrasekaran Padmapriyadarsini, Divyashri Jain, Shri Vijay Bala Yogendra Shivakumar, Kannan Thiruvengadam, Akshay N. Gupte, Beena Thomas, Aarti Kinikar, Krithika Sekar, Renu Bharadwaj, Chandra Kumar Dolla, Sanjay Gaikwad, S. Elilarasi, Rahul Lokhande, Devarajulu Reddy, Lakshmi Murali, Vandana Kulkarni, Neeta Pradhan, Luke Elizabeth Hanna, Sathyamurthi Pattabiraman, Rewa Kohli, Rani S., Nishi Suryavanshi, Shrinivasa B. M., Samyra R. Cox, Sriram Selvaraju, Nikhil Gupte, Vidya Mave, Amita Gupta, Robert C. Bollinger.

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
