## [Decision Letter · Decision Letter 0]

13 May 2020

PONE-D-20-08293

Tuberculosis preventive therapy should be considered for all household contacts of pulmonary tuberculosis patients in India

PLOS ONE

Dear Dr Paradkar,

Thank you for submitting your manuscript to PLOS ONE. After careful consideration, we feel that it has merit but does not fully meet PLOS ONE’s publication criteria as it currently stands. Therefore, we invite you to submit a revised version of the manuscript that addresses the points raised during the review process.

**In particular, Reviewer #2 raised a number of points that need to be addressed carefully and thoroughly in a revised manuscript.**

We would appreciate receiving your revised manuscript by Jun 27 2020 11:59PM. To enhance the reproducibility of your results, we recommend that if applicable you deposit your laboratory protocols in protocols.io, where a protocol can be assigned its own identifier (DOI) such that it can be cited independently in the future. For instructions see: http://journals.plos.org/plosone/s/submission-guidelines#loc-laboratory-protocols

We look forward to receiving your revised manuscript.

Kind regards,

Olivier Neyrolles

Academic Editor

PLOS ONE

Journal requirements

2. Please consider modifying your title to ensure that it is specific, descriptive, concise, and comprehensible to readers outside the field.

Reviewers' comments:

Reviewer's Responses to Questions

**Comments to the Author**

1. Is the manuscript technically sound, and do the data support the conclusions?

Reviewer #1: Yes

Reviewer #2: Partly

2. Has the statistical analysis been performed appropriately and rigorously? 

Reviewer #1: Yes

Reviewer #2: Yes

3. Have the authors made all data underlying the findings in their manuscript fully available?

Reviewer #1: Yes

Reviewer #2: No

4. Is the manuscript presented in an intelligible fashion and written in standard English?

Reviewer #1: Yes

Reviewer #2: No

5. Review Comments to the Author

Reviewer #1: It is a well-designed study that will add the evidence on LTBI treatment. It is also well written and analyzed

Abstract: No comment

Introduction: No comment

Methods: No comment

Results: No comment

Discussion: No comment

Reference: No comment

Reviewer #2: This is a well-designed prospective study that has tried to use mixed-effect Poisson regression for inferential data analyzed. The study has also included a high number of HHC to support the WHO recommendation. It is also a timely and relevant study in one of the high TB burden countries, INDIA.

General comments

This study has come up with an interesting and relevant finding that warrants a detailed and a bit extended discussion. Because the author is claiming/recommending TPT for all HHC and thus supporting the WHO recommendation. As compared to the findings, however, the discussion section is a bit brief and shorter. It seems there are other important findings that needs discussion; for example, why age, diabetes mellitus, and alcohol consumption are not related to iTBD? Could the way age (of HHC and index cases) was categorized, smoking and alcohol consumption were classified, and DM patients were presented logical and acceptable? This necessitates to revise the analysis section. There need to be a discussion as to why these are a not a factor for iTBD in this study, as compared to previous studies that verified as these are strong factors that are related to TB diseases, such as studies mentioned in references 2, 14, and 16. This can have importance in terms of prioritizing TPT in some low income countries who could not afford TPT for all HCC. This is a critical issue in need to be investigated. This is because, the authors are arguing and trying to convince strongly that no HHC is to be prioritized, no need to test for TBI prior to TPT provision. This requires answering the following questions.

1.Is it really feasible to provide TPT for all HHC considering limited logistics and availability of the newer combinations of drugs for TPT?

2.Which HHC to be given a priority?

Hence, the authors justification and arguments need to be a bit stronger and discussed in detail so that NTPs will be convinced to provide TPT for HHC without any prioritization and testing using TST or IGRA.

Besides, the way authors are listed and narrated, and their affiliation may be revised to align with the PLOS I format. The study could benefit from language revision; a bit longer and vague sentences are noted. The references as inserted in the body of the manuscript and listed in the reference need revision throughout. The formatting in the reference is not consistent and some are (E.g, reference # 5) incomplete. It is also worth considering the use of recent references; there are references older than 10 years (reference # 17, 21, 35, 36, 37, 44, and 48).

At end, the headings in lines 136, 147, 164,251 and 287 need to be revised and well narrated; at least, the author need to avoid the use of abbreviations/acronyms in the headings.

Specific comments

1. A sentence in Lines 32 and 33 is very important sentence but lacks clarity. In the same sentence, it is narrated that the study determined which HHC group are beneficial from TPT in India and other high TB burden country. Is that the real and specific objective of the study, as it was carried out among the Indian population?

2. TBD in line 35 should be fully written as it appeared for the first time.

3. A sentence in lines 33-36 could be revised to be narrated clearly and succinctly. As currently written, it is long and difficult to understand.

4. Line 59 & 60, the India’s contribution to the global TB incidence could be described in the form of proportion, 28% (2.8 of 10 million) or near to one-third of… Similarly, line 91 need to be revised.

5. Who were the children in your study (as related to age category)? Can we define them with reference?

6. A sentence in lines 108-110 is not clear and needs revision.

7. What was applied, oral or written informed consent? How was the consent of a child requested and obtained?

8. What are the specific psycho-social and medical history in lines 116 & 123, and household characteristics in line 132?

9. What is/are the reference (s) for the definition in lines 121-123 and 165-174?

10. A sentence in lines 126-131 is too long and better be narrated again. For example, presumptive TB cases should be defined, and which microbiological, tissue-based, or radiologically investigations are indicated for which specific signs or/and symptoms detected during the follow up?

11. Consider revision for a sentence in lines 141-143, seems two sentences.

12. A sentence in lines 148-149 lacks clarity.

13. On what basis was the age category made or classified as described in table 1?

14. Having TBD, and HHC without baseline TBI test the only exclusion criteria?

15. What IQR stands for in Line212-213?

16. Line 241-242 is not a result. The author may consider moving to data analysis section of the method.

17. The widowed and divorced ( as a sub-category of marital status) are with lower number to be categorized for the univariate and multivariate analysis. Consider categorizing these again.

18. What were the criteria to include the HHC characteristic to multivariate analysis? A lot of variables are not considered to be part of the cultivatable analysis, Table 1. For instance, p-value of 0.65 was included while p-value of 0.67 not included in the multivariate analysis. What does the sentence in lines 198-199 mean? These better be aligned with the way real analysis was made.

19. Where in the study have you applied the Fisher’s exact test and Wilcoxon rank sum tests? Because these were stated in lines 181-182.

20. How was the level of alcohol consumption and smoking level determined? If possible, objectively quantifying the degree of alcohol consumption and smoking is a better option. It seems that a fewer month’s period of smoking and alcohol consumption were lumped up with the heavy alcohol consumption and a longer period smoking. If the level or degree of smoking or alcohol consumption is not objectively defined, the relation or impact these have on the TBD is not well determined

21. Why Table 1 and 3 are written separately? Why not the author considers HHC characteristics in one go? The other option is that tables 1-3 could be presented in two forms; first table/s could be committed to the description of the HHC and index cases, and the second table/s could present the result of the univariate and multivariate analysis.

22. Check the consistency in the content of sentences in lines 66& 67, and lines 345 &346.

23. The sentence in lines 371-373 is critical yet needs revision to make it so clear. The way “resource’ is used makes the sentence a bit confusing.

6. PLOS authors have the option to publish the peer review history of their article (what does this mean?). If published, this will include your full peer review and any attached files.

Reviewer #1: Yes: Muluken Aseresa

Reviewer #2: No

---

## [Author Response · Author response to Decision Letter 0]

8 Jul 2020

Dated 8th July, 2020

Editor-in-Chief,

Plos One

Subject: #PONE-D-20-08293, Response to Reviewers for the manuscript titled, “Tuberculosis preventive therapy should be considered for all household contacts of pulmonary tuberculosis patients in India”. 

We thank you for the review of our manuscript. We have now revised the draft as per reviewer’s comments and have addressed each comment as below. We believe the changes suggested by the reviewers made the manuscript stronger and hope that you will consider our revised manuscript for publication.

Reviewer #1: No comments

Response: We appreciate the review and agreement with our manuscript.

Reviewer #2: This is a well-designed prospective study that has tried to use mixed-effect Poisson regression for inferential data analyzed. The study has also included a high number of HHC to support the WHO recommendation. It is also a timely and relevant study in one of the high TB burden countries, India.

Response: We appreciate your review of the manuscript and thank you for highlighting the importance of our findings.

1. General comments

a. Comment: This study has come up with an interesting and relevant finding that warrants a detailed and a bit extended discussion. Because the author is claiming/recommending TPT for all HHC and thus supporting the WHO recommendation. As compared to the findings, however, the discussion section is a bit brief and shorter. It seems there are other important findings that needs discussion; for example, why age, diabetes mellitus, and alcohol consumption are not related to iTBD? 

Response: We have updated the draft to expand the discussion to address the specific issues that the reviewer has raised below. 

b. Comment: Could the way age (of HHC and index cases) was categorized, smoking and alcohol consumption were classified, and DM patients were presented logical and acceptable? This necessitates to revise the analysis section. There need to be a discussion as to why these are a not a factor for iTBD in this study, as compared to previous studies that verified as these are strong factors that are related to TB diseases, such as studies mentioned in references 2, 14, and 16. 

Response: We thanks the reviewer for raising this very important issue with respect to the age, smoking, alcohol consumption and diabetes mellitus (DM) variables used in the analysis. 

i. We chose our age groups of the household contacts (HHC) based on the following: 1) <6 years is the cut-off for current Indian TB guidelines for TPT household contact recommendations in children. 2) 6-12 years was selected to include the older children. 3) 13-17 years represents the adolescents 4) 18-44 years to represent the younger adults and 5) ≥ 45 years was to represent the older adults. Also, we used age as a continuous variable in the Poisson regression analysis and did not find an association between age and iTBD.

ii. The data on smoking was collected under three mutually exclusive categories for the smoking variable, namely- current smoker, former smoker and non-smoker. However, there was only one former smoker who developed the outcome of incident TB disease (iTBD), while no current smoker developed iTBD. Therefore, in the original analysis presented in Table 1, we had combined these two categories as ‘Yes’ (any history of smoking) while the reference category being ‘No’ (no history of current or past smoking). Furthermore, there was no change in the inferences even when univariate and multivariate analysis were performed using the two smoking categories separately. Alternatively, as per the reviewer’s recommendation, to assess the impact of degree of smoking objectively and quantitatively, we also calculated the number of pack-years to reclassify the HHC in 3 categories of current smoker, former smoker and non-smoker and again separately performed the univariate and multivariate analysis (adult only model). This reclassification, however, did not change our conclusions in terms of association with iTBD. We have replaced the originally presented smoking variable with new smoking variable which is based on the pack-years analysis, in the revised Table 1. 

iii. We had also collected the ‘Alcohol Use Disorders Identification Test (AUDIT) score’, to quantitatively assess the alcohol dependence. Alcohol use disorder (AUD) was defined as having an AUDIT score of at least 8 points. However, there was no HHC with AUD who developed the outcome of incident TB disease (iTBD). Therefore, we have used the originally presented alcohol consumption variable in the revised Table 1. 

iv. Furthermore, we also revised the adult multivariate model replacing the original smoking variable with the new smoking variable described above (based on the pack-year analysis) however there was no change the in our conclusions in terms of association with iTBD. 

v. As defined in Table 1, footnote b, the standard definition of DM was used (known case of DM or, HB A1c ≥ 6.5%, or FBG ≥ 126 mg/dl or Random Blood glucose ≥ 200 mg/dl). The prevalence of DM among the at-risk population (adults) was as low as 9% (70 out of 734 adult HHC had DM), while only 1 of these 70 HHC with DM developed iTBD. 

In conclusion, no statistically significant association was found between any of the aforementioned variables (including age, smoking, alcohol consumption, DM) and iTBD. The lack of association with these risk factors in our study may possibly be due to- a) possible selection bias as this is not a population based study but a clustered HHC analysis, b) those with DM were more likely to be already diagnosed with prevalent TBD and therefore excluded from the analysis, c) Interaction of DM with malnutrition which is evident by DM having a non-significant protective effect due to higher BMI in diabetics, and d) due to the relatively low number of incident cases, we may not have power to identify individual risk factors that have relatively low prevalence in the HHC.

c. Comment: This can have importance in terms of prioritizing TPT in some low-income countries who could not afford TPT for all HHC. This is a critical issue in need to be investigated. This is because, the authors are arguing and trying to convince strongly that no HHC is to be prioritized, no need to test for TBI prior to TPT provision. This requires answering the following questions.

i. Is it really feasible to provide TPT for all HHC considering limited logistics and availability of the newer combinations of drugs for TPT?

ii. Comment: Which HHC to be given a priority? 

Hence, the authors justification and arguments need to be a bit stronger and discussed in detail so that NTPs will be convinced to provide TPT for HHC without any prioritization and testing using TST or IGRA.

Response: We agree with the reviewer’s comment about the importance of these issues. We felt this was a timely study, because the current WHO TPT recommendations and Indian guidelines are considering this as we described in lines 54-56, 66-67. The fact that we found that pre-screening test for identifying incident TB infection (iTBI) was not necessary to determine the eligibility for TB preventive treatment (TPT), makes provision of TPT for all HHC more feasible. 

The iTBI status itself was comprehensively assessed using all the possible definitions based on different published TST conversion and IGRA conversion cut off values, both in combinations and also by individual test result. However, even after using each of these iTBI definitions, we did not find any impact on iTBD.

As noted in the result section and in response to point 1.b. above, our findings suggest that with the exception of malnutrition and HIV positive status in the HHC, no other HHC subgroup is more likely to benefit from TPT than other HHCs. 

Furthermore, though some of the HHC subgroups like those with malnutrition and HIV infection, may benefit the most from TPT, nevertheless, the iTBD risk in HHC was much higher than what has been reported in the general population in India or South Africa. Therefore, being HHC of PTB patient itself is an important risk factor for iTBD and since our study did not find any predictors of disease progression, all HHC should be given TPT. 

Lastly, it might be programmatically easier to provide TPT to all HHC in the light of the fact that about 75% of the HHC in our study, meet a high risk criteria of either being a child < 6 years of age or among those ≥ 6 years of age with a positive TBI test or being HIV infected. However, we acknowledge that this is not a feasibility/cost-effectiveness study, so assessing the feasibility is beyond the scope of this analysis. Future feasibility/cost-effectiveness studies can help address this important consideration. 

To address these critical issues in above points 1.a., 1.b. and 1.c., we have revised the methods section (Lines 121-134), the results section (Lines 250, 299-301) and the discussion section (Lines 343-410), in the revised manuscript.

d. Comment: Besides, the way authors are listed and narrated, and their affiliation may be revised to align with the PLOS I format. 

Response: The author list and affiliations sequence have been revised and the appropriate symbols are used in the revised manuscript to align with the PLOS One format as per the reviewer’s comment.

e. Comment: The study could benefit from language revision; a bit longer and vague sentences are noted. 

Response: The manuscript has been revised thoroughly to address reviewer’s comment. 

f. Comment: The references as inserted in the body of the manuscript and listed in the reference need revision throughout. The formatting in the reference is not consistent and some are (E.g, reference # 5) incomplete. 

Response: The citations in the manuscript body and the reference list has been revised thoroughly to address the reviewer’s comment. 

g. Comment: It is also worth considering the use of recent references; there are references older than 10 years (reference # 17, 21, 35, 36, 37, 44, and 48).

Response: The original references 17 and 35 have been replaced with a recent reference as suggested by the reviewer. The original references 44 and 48 have been removed since there are more recent supporting references already cited for the relevant lines. The original references 35, 36, 37 are still relevant and form the basis for the TST induration cut offs used in the analysis, therefore, these references are retained in the revised manuscript. 

h. Comment: At end, the headings in lines 136, 147, 164,251 and 287 need to be revised and well narrated; at least, the author need to avoid the use of abbreviations/acronyms in the headings.

Response: The headings in the lines aforementioned by the reviewer are appropriately narrated and the abbreviations have been replaced with full forms, in lines 154, 165, 182, 269, and 305 of the revised manuscript. 

2. Specific comments

a. Comment: A sentence in Lines 32 and 33 is very important sentence but lacks clarity. In the same sentence, it is narrated that the study determined which HHC group are beneficial from TPT in India and other high TB burden country. Is that the real and specific objective of the study, as it was carried out among the Indian population?

Response: This has been clarified in the introduction by removing the phrase “other high TB burden country”, in lines 32-34 in the revised manuscript. 

b. Comment: TBD in line 35 should be fully written as it appeared for the first time.

Response: In the revised manuscript, TB disease (TBD) is written fully in line 34, where it has been mentioned for the first time. 

c. Comment: A sentence in lines 33-36 could be revised to be narrated clearly and succinctly. As currently written, it is long and difficult to understand.

Response: The original sentence has been simplified in multiple sentences in the lines 34-36 of the revised manuscript, for the purpose of clarity.

d. Comment: Line 59 & 60, the India’s contribution to the global TB incidence could be described in the form of proportion, 28% (2.8 of 10 million) or near to one-third of… Similarly, line 91 need to be revised.

Response: In the revised manuscript, the global TB incidence is described in the form of proportion in lines 61-62 and the original sentence in line 91 has been deleted. 

e. Comment: Who were the children in your study (as related to age category)? Can we define them with reference?

Response: As mentioned in the point 1.b. above, broadly, HHC < 18 years of age (non-adult HHC), including the younger children, the older children and the adolescents, are referred to as children. In India, ≥ 18 years is legally the age when a minor individual attains adulthood. 

f. Comment: A sentence in lines 108-110 is not clear and needs revision.

Response: The relevant sentence has been simplified for clarity, in lines 110-112 of the revised manuscript.

g. Comment: What was applied, oral or written informed consent? How was the consent of a child requested and obtained?

Response: The written informed consent was obtained from the participating adult HHC (≥ 18 years of age) and from the legal guardian if the participating HHC was a child < 18 years of age. As per the local IRB norms a written informed assent was sought and obtained from children within the age group of ≥ 8 to < 18 years. This has been clarified in lines 112-117 of the revised manuscript. 

h. Comment: What are the specific psycho-social and medical history in lines 116 & 123, and household characteristics in line 132?

Response: We have clarified the data variables referred to under the psychosocial, medical history and household characteristics, in lines 122-135, 150-151 of the revised manuscript. 

i. Comment: What is/are the reference (s) for the definition in lines 121-123 and 165-174?

Response: The relevant references have been added for the definitions of undernutrition and TBD in lines 136-139 and lines 184-192 of the revised manuscript, respectively. 

j. Comment: A sentence in lines 126-131 is too long and better be narrated again. For example, presumptive TB cases should be defined, and which microbiological, tissue-based, or radiologically investigations are indicated for which specific signs or/and symptoms detected during the follow up?

Response: The sentence has been simplified by splitting in two sentences in lines 142-149 of the revised manuscript. Also, symptoms and corresponding investigations have been clarified as suggested by the reviewer.

k. Comment: Consider revision for a sentence in lines 141-143, seems two sentences.

Response: The sentence has been simplified by splitting in two separate sentences in lines 160-162 of the revised manuscript.

l. Comment: A sentence in lines 148-149 lacks clarity.

Response: The sentence has been simplified for clarity in lines 167-168 of the revised manuscript.

m. Comment: On what basis was the age category made or classified as described in table 1?

Response: As mentioned above in point 1.b., the age groups were classified based on the following: 1) <6 years is the cut-off for current Indian TB guidelines for TPT household contact recommendations in children. 2) 6-12 years was selected to include the older children. 3) 13-17 years represents the adolescents 4) 18-44 years to represent the younger adults and 5) ≥ 45 years was to represent the older adults. 

n. Comment: Having TBD, and HHC without baseline TBI test the only exclusion criteria?

Response: To ensure that we include only those HHC in the analysis who were at risk of TBD during the study follow up, we excluded HHC with TBD diagnosed at baseline. To identify the TBI test (TST and/or IGRA) conversion and its impact on the incident TBD, availability of at least one test result (TST or IGRA) at baseline was essential, therefore the HHC with no baseline TBI test results for both the tests were excluded. There were the only two exclusion criteria applied, as depicted in Figure 1. 

o. Comment: What IQR stands for in Line 212-213?

Response: IQR stands for interquartile range. This full form has been added at the first mention in line 233, in the revised manuscript.

p. Comment: Line 241-242 is not a result. The author may consider moving to data analysis section of the method.

Response: The sentence has been moved to the data analysis section, in lines 198-200 of the revised manuscript. 

q. Comment: The widowed and divorced as a sub-category of marital status are with lower number to be categorized for the univariate and multivariate analysis. Consider categorizing these again.

Response: The widowed and divorced categories have been merged in Table 1 of the revised manuscript. The original findings still do not change after this re-categorization as there were no HHC from this category who developed iTBD. 

r. Comment: What were the criteria to include the HHC characteristic to multivariate analysis? A lot of variables are not considered to be part of the multivariable analysis, Table 1. For instance, p-value of 0.65 was included while p-value of 0.67 not included in the multivariate analysis. What does the sentence in lines 198-199 mean? These better be aligned with the way real analysis was made.

Response: The sentence in original lines 198-199 means that, those HHC characteristics which were found to be associated with iTBD in the univariate analysis were included in the overall model and/or the adult multivariate models, as relevant. Additionally, those HHC characteristics that were not statistically significant in the univariate analysis but known to be the published risk factors for iTBD, were included in the multivariate model. This has been clarified in lines 215-219 of the revised manuscript.

s. Comment: Where in the study have you applied the Fisher’s exact test and Wilcoxon rank sum tests? Because these were stated in lines 181-182.

Response: We agree with the reviewer that Fisher’s exact and Wilcoxon rank sum tests were not applied in the final analysis presented. Therefore, the relevant sentence has been deleted.

t. Comment: How was the level of alcohol consumption and smoking level determined? If possible, objectively quantifying the degree of alcohol consumption and smoking is a better option. It seems that a fewer month’s period of smoking and alcohol consumption were lumped up with the heavy alcohol consumption and a longer period smoking. If the level or degree of smoking or alcohol consumption is not objectively defined, the relation or impact these have on the TBD is not well determined.

Response: Please refer to our detailed response to point 1.b. above, which addresses the impact of objective quantification of smoking and alcohol variables on the association with iTBD.

u. Comment: Why Table 1 and 3 are written separately? Why not the author considers HHC characteristics in one go? The other option is that tables 1-3 could be presented in two forms; first table/s could be committed to the description of the HHC and index cases, and the second table/s could present the result of the univariate and multivariate analysis.

Response: We appreciate the reviewer’s suggestion to either have tables 1-3 merged or present them as two tables instead of three separate tables, but that would make the tables very lengthy. However, we are happy to revise the tables as per the editor’s preference. 

v. Comment: Check the consistency in the content of sentences in lines 66& 67, and lines 345 &346.

Response: These sentences in lines 66-67 and lines 345-346 (line numbers as per the original manuscript) are both different. Line 66-67 mentions that revision of Indian guidelines for provision of TPT is currently under consideration, however are not yet revised. Line 345-346 states the current guideline regarding provision of TPT. 

w. Comment: The sentence in lines 371-373 is critical yet needs revision to make it so clear. The way “resource’ is used makes the sentence a bit confusing.

Response: This critical sentence in the discussion section (Lines 408-411) is revised as follows, “In summary, our study supports the new WHO guidelines to rapidly screen all HHC of PTB patients and to offer TPT to all HHC without TBD and do not suggest any clear benefit of TBI testing at baseline or during follow-up to further risk stratify recently-exposed HHC for targeted TPT.” 

Journal requirements

Response: The manuscript has been revised and the files have been renamed, to meet the PLOS ONE’s style requirements. 

2. Please consider modifying your title to ensure that it is specific, descriptive, concise, and comprehensible to readers outside the field.

Response: The title has been revised as follows, “Tuberculosis preventive treatment should be considered for all household contacts of pulmonary tuberculosis patients in India”

Response: The data from this study are part of a large multisite consortium and can be made available with use of a data sharing agreement as per Indian government norms. Specific requests can be placed through the non-author institutional point of contact as follows: Sameer Khan, Data Manager, BJ Government Medical College Johns Hopkins University Clinical Research Site (sameeriz@hotmail.com). This information has been added to the revised cover letter. 

Thank you for the opportunity to submit our manuscript for consideration. Please do not hesitate to contact us with questions.

Sincerely,

Mandar Paradkar, MBBS, DCH, MPH

Corresponding author

Email: drman23@gmail.com

---

## [Decision Letter · Decision Letter 1]

14 Jul 2020

Tuberculosis preventive treatment should be considered for all household contacts of pulmonary tuberculosis patients in India

PONE-D-20-08293R1

Dear Dr. Paradkar,

We’re pleased to inform you that your manuscript has been judged scientifically suitable for publication and will be formally accepted for publication once it meets all outstanding technical requirements.

Kind regards,

Olivier Neyrolles

Section Editor

PLOS ONE

Reviewers' comments:

Reviewer's Responses to Questions

**Comments to the Author**

1. If the authors have adequately addressed your comments raised in a previous round of review and you feel that this manuscript is now acceptable for publication, you may indicate that here to bypass the “Comments to the Author” section, enter your conflict of interest statement in the “Confidential to Editor” section, and submit your "Accept" recommendation.

Reviewer #2: All comments have been addressed

2. Is the manuscript technically sound, and do the data support the conclusions?

Reviewer #2: Yes

3. Has the statistical analysis been performed appropriately and rigorously? 

Reviewer #2: Yes

4. Have the authors made all data underlying the findings in their manuscript fully available?

Reviewer #2: Yes

5. Is the manuscript presented in an intelligible fashion and written in standard English?

Reviewer #2: Yes

6. Review Comments to the Author

Reviewer #2: (No Response)

7. PLOS authors have the option to publish the peer review history of their article (what does this mean?). If published, this will include your full peer review and any attached files.

Reviewer #2: **Yes: **Zewdu Gashu Dememew

---

## [Editor Report · Acceptance letter]

16 Jul 2020

PONE-D-20-08293R1 

Tuberculosis preventive treatment should be considered for all household contacts of pulmonary tuberculosis patients in India 

Dear Dr. Paradkar:

I'm pleased to inform you that your manuscript has been deemed suitable for publication in PLOS ONE. Congratulations! Your manuscript is now with our production department. 

Kind regards, 

on behalf of

Dr. Olivier Neyrolles 

Section Editor

PLOS ONE